# Stress distribution on different bar materials in implant-retained palatal obturator

**Regina Furbino Villefort**[1‡], **João Paulo Mendes Tribst**[2‡*], **Amanda Maria de Oliveira Dal Piva**[2☉], **Alexandre Luiz Borges**[2☉], **Nívia Castro Binda**[1☉], **Carlos Eduardo de Almeida Ferreira**[3☉], **Marco Antonio Bottino**[2☉], **Sandra Lúcia Ventorim von Zeidler**[1☉]

**1** Biotechnology Program, Federal University of Espírito Santo, Rede Nordeste de Biotecnologia (RENORBIO), Vitória, Espírito Santo, Brazil, **2** Post-Graduate Program in Restorative Dentistry (Prosthodontic), Department of Dental Materials and Prosthodontics, Institute of Science and Technology, São Paulo State University (Unesp/SJC), São José dos Campos, SP, Brazil, **3** Private Practitioner, Vitória, Espírito Santo, Brazil

☉ These authors contributed equally to this work.
‡ These authors also contributed equally to this work.
* Joao.tribst@gmail.com

**Data Availability Statement:** All relevant data are within the manuscript and its Supporting Information files.

## Abstract

Implant-retained custom-milled framework enhances the stability of palatal obturator prostheses. Therefore, to evaluate the mechanical response of implant-retained obturator prostheses with bar-clip attachment and milled bars, in three different materials under two load incidences were simulated. A maxilla model which Type IIb maxillary defect received five external hexagon implants (4.1 x 10 mm). An implant-supported palatal obturator prosthesis was simulated in three different materials: polyetheretherketone (PEEK), titanium (Ti:90%, Al:6%, V:4%) and Co-Cr (Co:60.6%, Cr:31.5%, Mo:6%) alloys. The model was imported into the analysis software and divided into a mesh composed of nodes and tetrahedral elements. Each material was assumed isotropic, elastic and homogeneous and all contacts were considered ideal. The bone was fixed and the load was applied in two different regions for each material: at the palatal face (cingulum area) of the central incisors (100 N magnitude at 45°); and at the occlusal surface of the first left molar (150 N magnitude normal to the surface). The microstrain and von-Mises stress were selected as criteria for analysis. The posterior load showed a higher strain concentration in the posterior peri-implant tissue, near the load application side for cortical and cancellous bone, regardless the simulated material. The anterior load showed a lower strain concentration with reduced magnitude and more implants involving in the load dissipation. The stress peak was calculated during posterior loading, which 77.7 MPa in the prosthetic screws and 2,686 με microstrain in the cortical bone. For bone tissue and bar, the material stiffness was inversely proportional to the calculated microstrain and stress. However, for the prosthetic screws and implants the PEEK showed higher stress concentration than the other materials. PEEK showed a promising behavior for the bone tissue and for the integrity of the bar and bar-clip attachments. However, the stress concentration in the prosthetic screws may represent an increase in failure risk. The use of Co-Cr alloy can reduce the stress in the prosthetic screw; however, it increases the bone strain; while the Titanium showed an intermediate behavior.

**Funding:** This research was supported by FAPES/CAPES Grant: FAPES/CAPES N° 10/2018 – PROFIX 2018 Process number: 83574662 Funding granted to: Regina Furbino Villefort The funders had no role in study design, data collection and analysis, decision to publish, or preparation of the manuscript. There was no additional external funding received for this study.

**Competing interests:** The authors have declared that no competing interests exist.

## Introduction

Palatal obturator prostheses and microvascular reconstructive techniques are common treatment options for rehabilitation of those patients undergoing maxillectomy during surgical resection of tumors [1, 2]. Nevertheless, prosthetic rehabilitation remains the most widely used approach, especially for large maxillary defects [3, 4], with improvements in oral functions [5] and significant increase in the quality of life of patients [6]. The design of the obturator prosthesis and retention mechanisms depend basically on the size and location of the defect, clinical conditions of bone, remaining teeth, soft tissue, muscle control, and the physical and mental health conditions of the patient [7]. However, the retention of these prostheses for edentulous patients is often insufficient. In a systematic review on functional outcomes in oncologic patients, the authors observed that rehabilitation with implants significantly improved prosthesis retention, presenting a beneficial effect for masticatory efficiency and greater satisfaction of patients [8].

The possibility of using computer-aided design/computer-aided-manufacturing (CAD/CAM) technology to machine reliable prostheses in different materials (metal, ceramics and polymers) has diversified the standard designs of implant-supported prostheses and their clinical performance [9, 10]. Furthermore, CAD/CAM frameworks retained by implants improved the palatal obturator stability and functional results for patients with partial maxillectomy [11]. Titanium and cobalt-chromium alloys represent standard materials for CAD/CAM frameworks, with good performance and similar fit [12], while polyetheretherketone (PEEK) is an inert, non-allergenic polymeric biomaterial, indicated as a substitute for metal alloys in assorted types of prostheses and orthoses, including craniofacial prostheses [13, 14].

*In vitro* studies and short-term clinical reports evaluated the use of PEEK in dentistry for partial/total; fixed/removable; tooth supported/implant-supported [15–21], and maxillofacial prosthesis, including palatal obturators [22]. PEEK has shown some advantages such us the fact of it has an elastic modulus similar to the native bone, is easily obtained in personalized three-dimensional (3D) forms, propitiates the manufacture of radiolucent prostheses, with good biomechanical properties, and less accumulation of biofilm than ceramics and metallic alloys, which are usual materials in restorative dentistry [15–21]. Despite that, the biomechanical behavior of PEEK obturator prostheses retained by implants remains unknown, as well the mechanical response during chewing loads in the implants, prosthetics screws, cortical and cancellous bone for the patients rehabilitated with implant-retained obturator prostheses.

The assessment of biomechanical behavior can be performed through simulations to obtain pre-clinical data with bioengineering tools such as strain gauge, digital image correlation, photoelasticity and finite element analysis (FEA). The latter allows us to understand how the distribution of strain in bone tissue and stress in implants can be influenced by the restorative material [23], prosthesis and framework design [24, 25], manufacturing technique [26], number and distribution of implants [27–29] and attachment systems [30, 31]. Computer-assisted implant planning has become an important diagnostic and therapeutic tool in modern dentistry. The ideal implant positions can be planned virtually with the help of guided surgery software allowing for three-dimensional visualization before treatment. The combination of planning and case study by FEA can help to choose and predict the most suitable mechanical results.

Thus, the objective of this study was to evaluate the mechanical response of implant-retained obturator prostheses with bar-clip attachment and milled bars in three different materials: PEEK, titanium and cobalt-chromium alloys in different load incidences. The null hypothesis is that different materials for the framework will not modify the mechanical response in the analyzed structures regardless of the applied load.

## Methods

### Pre-processing

A computer tomography (CT) from São Paulo State University database, without maxillofacial abnormalities, were saved in DICOM (Digital Imaging and Communications in Medicine) format. This file comes from the University database, the authors have no access to any identifying information or taking the CT scan. The DICOM file was converted to STL (stereolithography) file in a 3D slicer software. Using CAD (computer-aided design) software (Rhinoceros Version 4.0 SR8, McNeel North America, Seattle, WA, USA), a model of an edentulous maxilla was constructed following the main anatomical characteristics of the patient's bone: size, shape and absence of lesion. The next step was to reconstruct the NURBS (non-uniform rational B-spline) surfaces from mesh with precision. For that, the BioCAD method [32] was applied and the anatomical lines of the surface were created. The pre-processing phase is summarized in a flowchart (Fig 1). The 3D volumetric model of the bone was then finished based on the surface created by the curve network manually generated (Fig 2A). The cortical bone (Fig 2E) contained 1 mm thickness in juxtaposition with cancellous bone (Fig 2D) [33]. The command offset surface was used to create the soft tissue with 2 mm thickness [34] (Fig 2F).

External hexagon implants (10 × 4.1 mm), previously modeled [25] were selected. The platform had a diameter of 4.1 mm, similar to a regular conventional implant. The external hexagon was extruded (0.7 mm high) and attached to the previously created cylindrical body [33] (Fig 2H). The minimum distance between the implants was 4 mm (Fig 2G). The prosthetic screw was modeled for each implant (Fig 2I). The total number of implants and their position were based in a previous report with the similar prosthetic approach [11] (Fig 2B). After that Type IIb defect [35] was simulated following the length and width from a clinical report [36] (Fig 2B).

The bar was modeled following the maxilla shape and the implants position. It presents 3 mm maximum thickness and 4 mm width, rounded corners and flat surfaces. An anteroposterior structure was used crossing the hemimaxillectomy defects similar to [11] report (Fig 3).

After that, the full-arch total prosthesis was modeled containing artificial teeth [33] and palatal coverage (Fig 2J). The clip connector for the fixture system was modeled in the same position as the bar from the framework.

### Processing

For the FEA, each solid geometry was imported to the computer aided engineering (CAE) software (ANSYS 19.2, ANSYS Inc., Houston, TX, USA) in STEP format. A 3D mesh was generated, and tetrahedral elements were considered for the models. A convergence test (maximum of 10% of difference in results between each calculation) determined the total number of elements (169,546) and nodes (306,914) for the model (Fig 4A). The Elastic modulus and Poisson ratio of each material were assigned to each solid component with isotropic and homogeneous behavior (Table 1). The contacts were considered perfectly bonded between the structures.

For the boundary conditions, the bottom surface of cancellous bone was restricted in all directions Fig 4B). Wedel et al. [44] established 120N as the occlusal force in patients with congenital and acquired maxillofacial defects while another studies reported the load range of 48–300 N for overdenture patients [45–47]. In this study, both simulated loads are inside this range.

The load was applied in two different moments for each material: at the palatal face (cingulum area) of the central incisors with 100 N magnitude applied at 45˚ [48] (Fig 4C); and at the occlusal surface of the first left molar with 150 N magnitude applied normal to the surface [29] (Fig 4D).

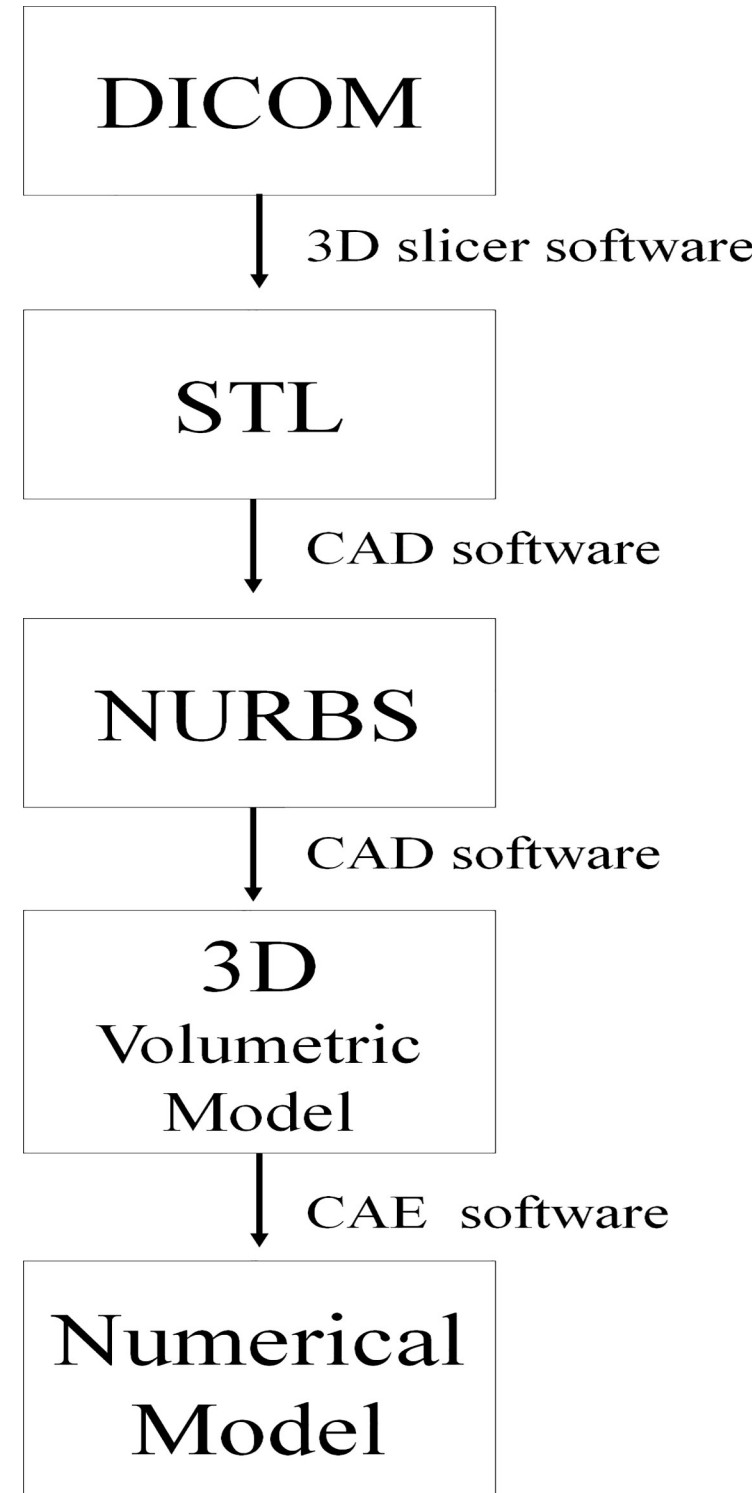

**Fig 1. Process flowchart.** Diagrammatic representation of the steps and the correlated software applied in the preprocessing and processing phases.

**Fig 2. 3D model and geometries.** Edentulous maxilla (A); Type IIb maxillary defect (B); Implants distribution on maxillary crestal bone (C); Cancellous bone (D); Cortical bone (E); Soft tissue (F); Five external hexagon implants (G); Bar and bar-clip attachments (H); Five prosthetic screws (I); Full-arch total prosthesis (J).

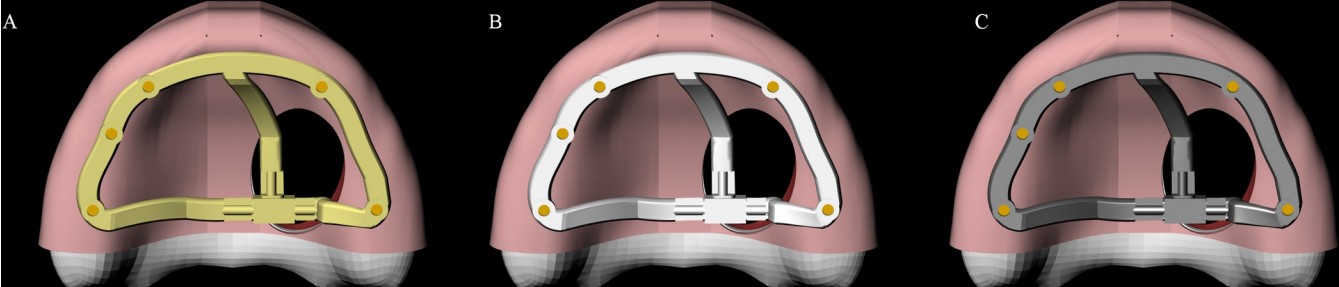

**Fig 3. 3D model.** Milled implant-retained bar with 3 clip-bar attachments in different materials. PEEK (A); Titanium alloy (B); Cobalt-Chromium alloy (C).

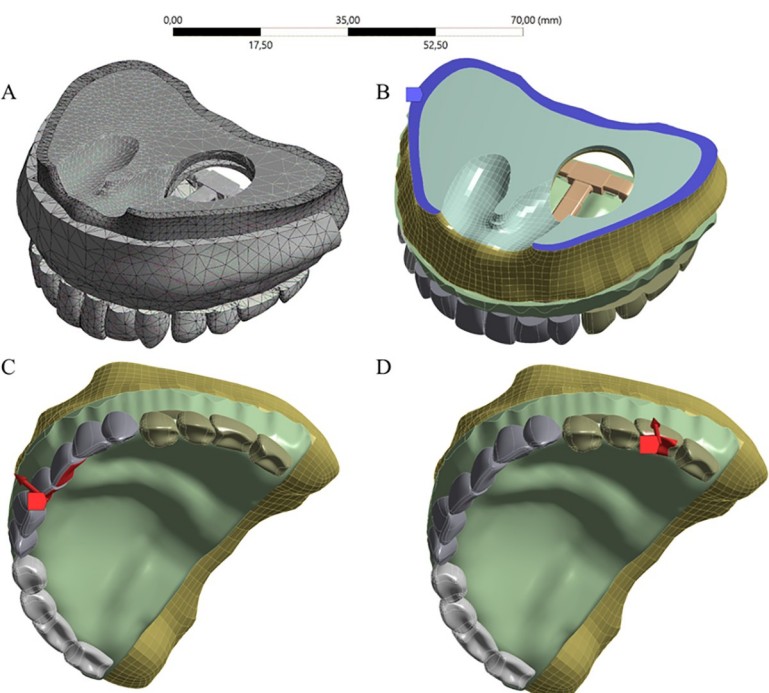

**Fig 4. Boundary conditions and loadings configuration in FEA models.** Mesh (A); Boundary conditions (B); Anterior load was applied at cingulum area of the central incisor (C); Posterior load was applied at occlusal surface of the first molar (D).

Results were reported in von Mises stress [25] distribution for the framework, implants and screw; and in microstrains (με) for bone tissue [49].

## Results

The calculated microstrain distribution in the maxilla as a function of the framework's material and load incidence were plotted in colorimetric graphs in the Figs 5 and 6 for cortical and cancellous bone, respectively. It was possible to observe that the posterior load showed a higher strain concentration in the posterior peri-implant tissue, near the load application side for cortical and cancellous bone. The anterior load showed a lower strain concentration with reduced magnitude and more implants involving in the load dissipation.

Moreover, the higher the framework elastic modulus, the higher the bone strain regardless the evaluated bone tissue and load incidence. The peak value of each group was exported from the analysis software to quantify the strain (Table 2). According to Wolff's law, strain values

**Table 1. Mechanical properties of the materials/solid geometry used in the current study.**

| Material/solid geometry | Young's Modulus (GPa) | Poisson Ratio |
|---|---|---|
| Cancellous bone [37] | 1.37 | 0.3 |
| Cortical bone [38] | 14.7 | 0.3 |
| Soft tissue [39] | 0.68 | 0.45 |
| PEEK [40] | 3.7 | 0.4 |
| Titanium alloy [41] | 110 | 0.3 |
| Cobalt-Chromium alloy [42] | 200 | 0.3 |
| Acrylic resin [43] | 2.83 | 0.45 |

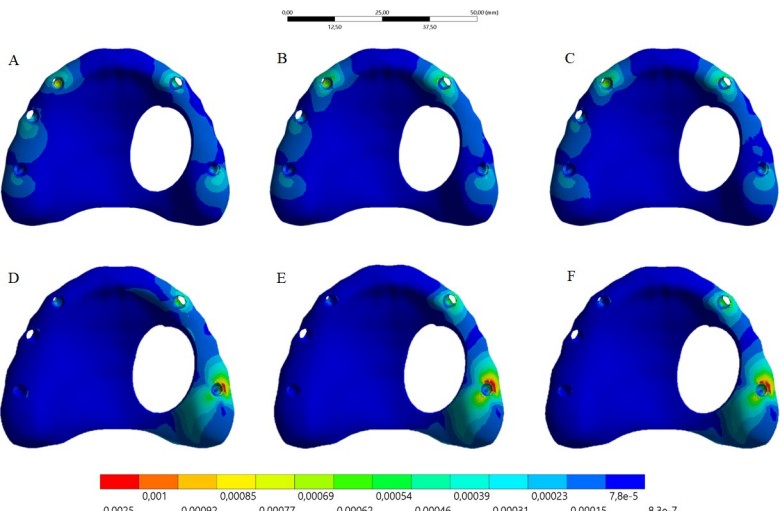

**Fig 5. Microstrain distribution in the maxillary cancellous bone.** Upper line: anterior loading; Bottom line: posterior loading. Framework's material: PEEK (A and D); Titanium alloy (B and E); Cobalt-Chromium alloy (C and F).

below 50 mm/mm are able to promote bone remodeling by disuse, and those values above 3000 mm/mm are able to promote bone remodeling by micro-damage. Thus, the three types of framework's material showed no values able to induce an unwanted bone remodeling.

The stress distribution in the framework for all groups is displayed in the Fig 7. Similar to the bone tissue mechanical behavior, the higher the framework elastic modulus, the higher the stress concentration regardless the load incidence. For posterior load, the stress concentration occurred in the lateral side; and for the anterior load, the stress concentration occurred between the anterior implants, near the screw access holes.

A higher stress concentration in the framework promoted a lower stress concentration in the implant (Figs 8 and 9) and prosthetic screw (Figs 10 and 11). Observing the results displayed in the Fig 9 it is possible to see that the higher the framework elastic modulus, the lower

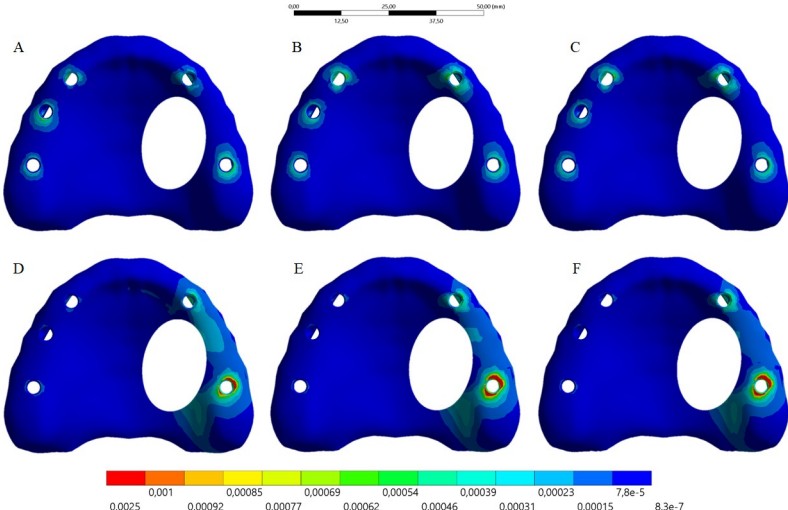

**Fig 6. Microstrain distribution on the maxillary cortical bone.** Upper line: anterior loading; Bottom line: posterior loading. Framework's material: PEEK (A and D); Titanium alloy (B and E); Cobalt-Chromium alloy (C and F).

**Table 2. Results in terms of bone microstrain (με) and stress peak values (MPa) according to the framework's material and load incidence.**

| | PEEK | | Titanium | | CoCr | |
|---|---|---|---|---|---|---|
| | Anterior | Posterior | Anterior | Posterior | Anterior | Posterior |
| Cortical Bone (με) | 411 | 1,460 | 460 | 2,360 | 492 | 2,686 |
| Cancellous Bone (με) | 588 | 1,258 | 567 | 1,896 | 587 | 1,987 |
| Framework (MPa) | 4.6 | 9.5 | 66.3 | 51.4 | 71.7 | 62.5 |
| Implant (MPa) | 25.4 | 86.4 | 20.8 | 79.2 | 19.8 | 74.1 |
| Prosthetic Screw (MPa) | 53.2 | 77.7 | 22.6 | 40.3 | 22.1 | 25.3 |

the stress concentration in the implants. The posterior load showed a higher stress magnitude with more red fringes in the colorimetric stress map in comparison with the anterior load, with the most posterior implant being the most affected. For the prosthetic screw, the same stress pattern observed in the implants occurred (Fig 11). The highest stress concentration calculated for the screw is the combination of PEEK framework and posterior load.

The results in terms of stress peak values (MPa) in the framework, prosthetic screw, and implant are summarized in Table 2.

## Discussion

This study aimed to evaluate the mechanical response of implant-retained obturator prostheses with clip-bar system, in different materials. The hypothesis was rejected due to the differences of stress concentration observed among materials.

Technical and biological complications or even failures are usual outcomes in dental prosthesis and it should be considered, in order to prevent them. Biomechanical tests are consolidated methods which have been contributing to the comprehension of the behavior of restorative materials and this study compared the mechanical response of CoCr and titanium alloys (materials commonly used for CAD/CAM frameworks) with the PEEK, an alternative to metal alloys in dentistry. The elastic modulus of Co-Cr alloy (200 GPa) [42] is almost double of the titanium's modulus (110 GPa) [41] and both have the same Poisson coefficient, while PEEK presents both, elastic modulus and Poisson coefficient, closer to those of acrylic resin

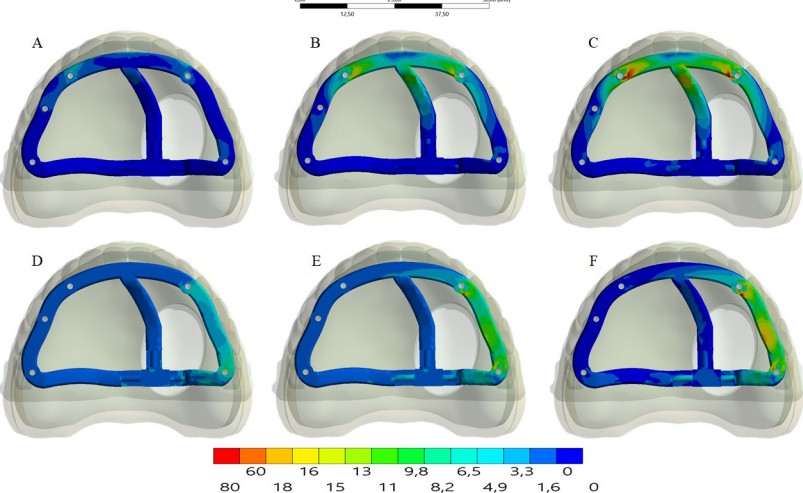

**Fig 7. The stress distribution in the framework.** Upper line: anterior loading; Bottom line: posterior loading. Framework's material: PEEK (A and D); Titanium alloy (B and E); Cobalt-Chromium alloy (C and F).

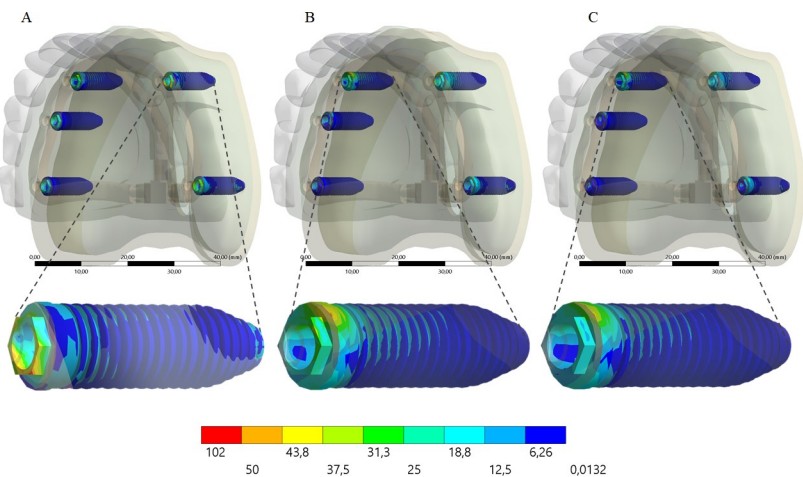

**Fig 8. Maps of von Mises stress distribution results for implants according to framework's material, under anterior loading.** PEEK (A); Titanium (B); Cobalt-Chromium alloy (C). On the bottom line, an enlarged view of the implants that presented the highest stress concentration.

[43] (material used in synthetic teeth). This difference makes the metal alloys behave differently from the PEEK when the anterior and posterior loads are applied. Therefore, the higher stress concentration was calculated in the CoCr bar, followed by titanium and finally the PEEK bar.

According to Bhering et al. [50], rigid frameworks transmit lower load to the implant and prosthetic components, when compared to the less rigid ones. Nonetheless, variations of framework material rigidity did not demonstrate a significant effect on the stress values in the marginal bone around the implants [51] and Medeiros et al. [26] observed that the occlusal coating had a greater influence on the load dissipations than the framework. In consonance with Erkmen et al. [52] the current study demonstrated that by using less rigid material for milled bar in implant-retained prostheses the stresses within both, the framework and the veneering parts, decreased due to the flexible nature of the material that absorbs stresses. The

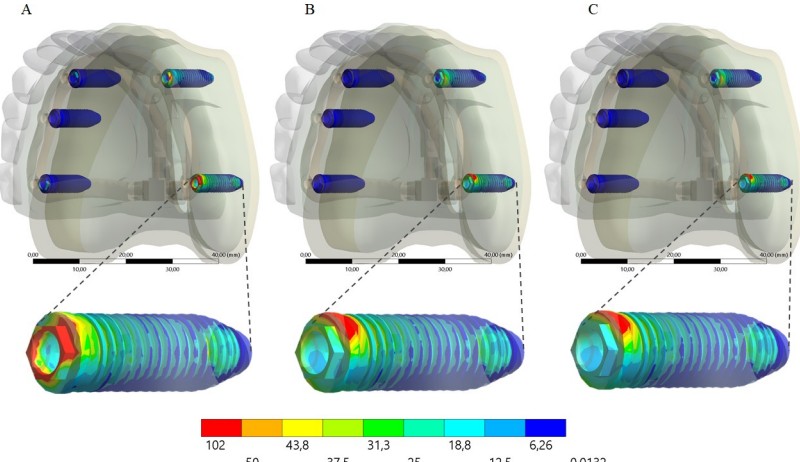

**Fig 9. Maps of von Mises stress distribution results for implants according to framework's material, under posterior loading.** PEEK (A); Titanium (B); Cobalt-Chromium alloy (C). On the bottom line, an enlarged view of the implants that presented the highest stress concentration.

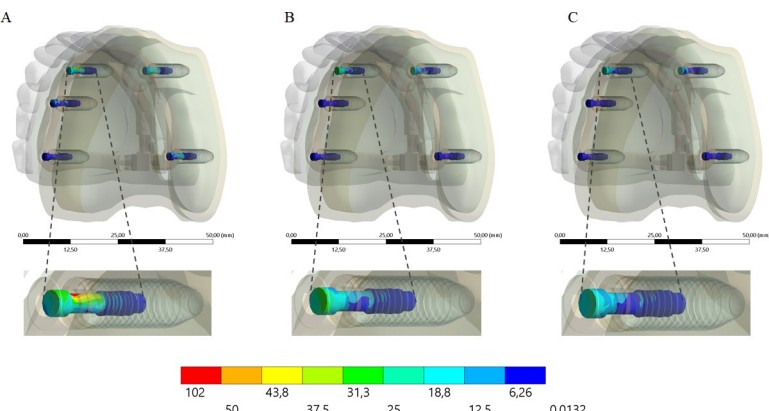

**Fig 10. Maps of von Mises stress distribution results for prosthetic screw according to framework's material, under anterior loading.** PEEK (A); Titanium alloy (B); Cobalt-Chromium alloy (C). On the bottom line, an enlarged view of the prosthetic screws that presented the highest stress concentration.

results from a longitudinal study on the use of PEEK milled bar as framework for implant-supported full-arch fixed prostheses suggest that this material may become an appropriate treatment option [21].

Apart from the material, the custom-milled framework design also influences the stress distribution [25]. In conventional maxillary overdentures that do not require palatal coverage, the stresses tend to be concentrated in the distal of the last implant, in the cantilever region [30]. However, obturator prostheses aims both coverage and adequate sealing of the oroantral communication, and thus the cantilever ceases to exist if there is residual bone on the affected side, or if zygomatic implants are installed. The maximum displacement of the obturator prosthesis increases as less residual bone is present, as well as less implants and clips [53]. Therefore, the unaffected side by maxillectomy should receive a larger number of implants to reduce the stress concentration in the framework [27]. The present study evaluated the performance of a milled bar for obturator prosthesis with polygonal geometry, without cantilever and with total palatal coverage and three clips in the region of the maxillary defect, supported by five implants. This model was based on a clinical case [36] and it was observed that despite the number of implants at unaffected side, the unilateral posterior loading promoted higher stress

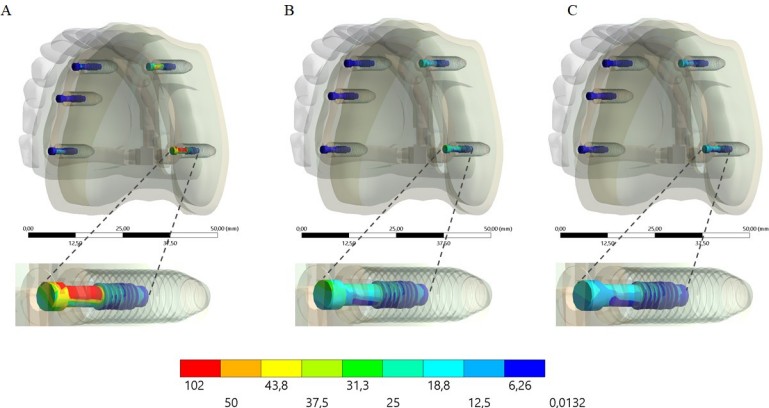

**Fig 11. Maps of von Mises stress distribution results for prosthetic screw according to framework's material, under posterior loading.** PEEK (A); Titanium alloy (B); Cobalt-Chromium alloy (C). On the bottom line, an enlarged view of the prosthetic screws that presented the highest stress concentration.

concentration in the mesial buccal and distal palatal portions of the cervical bone (cancellous and cortical) on the most distal implant on the affected side.

Regarding to the anchorage system, bar-clips possess a more favorable design to distribute the loads than O-rings [54] and presents lower strain values when submitted to compressive occlusal loads [55]. Furthermore, this system had a better biomechanical performance with the lowest strain values around the dental implants when subjected to forces simulating prosthesis removal [56]. In the current study, three the bar-clips were simulated in different materials and it was observed that PEEK attachments concentrated less stress than the metallic ones. This could be consider a possible advantage of using PEEK, taking into account that according to Tanoue et al. [57] clips can prevent the fracture of the prosthesis base more than metal clips, regardless of the number, due the lower concentration of stress observed around plastic clips. However, it is important to emphasize that the most frequent complication in implant-supported overdentures with bar-clip system are associated with the retention clips, requiring its replacement in 33% of cases [58].

For implant-supported rehabilitation, the marginal bone loss with exposure of the implant threads can be considered one of the biological complications. Frost [49] suggests that unrepaired bone resorption starts when strains exceed 3000μ strain. In the present study, the use of PEEK milled bar suggests a better mechanical performance for bone tissue with less possibility of unwanted bone resorption due to less peri-implant deformation independent on the simulated masticatory load.

Stress concentration on prosthetic screws is influenced by implant connection and the material selected for abutments and frameworks. Less rigid abutments like those manufactured in reinforced fiber composite and PEEK are not absolutely relevant for morse-taper implant [59]. In contrast, for external hexagon implants, flexible prosthetic frameworks increases the stress generated in the prosthetic screw threads [23] and may decrease the survival of restorations under cyclic fatigue [60]. This is relevant because abutment screw loosening, the fixing screw fractures, screw retightening and loosening of the abutment are usual technical complications in implant-retained prosthesis. Thus, the results of the present study suggest that the use of PEEK might facilitate the emergence of mechanical complications in prosthetic screws compared with metal frameworks.

Inherent limitations of the finite element analysis studies and biologic simulations were observed in the present investigation and two assumptions figured as the principals. The first one was the simulated materials presented a homogeneous structure and linearly isotropic behavior that do not represent the defects population incorporated during the prosthesis manufacturing. The second was that the implants were assumed 100% osseointegrated, although histomorphometric studies indicated that there is not a 100% bone–implant interface. Nonetheless, these assumptions are consistent with other FEA studies [29, 50, 52] and are consequences of the challenges in establishing the properties of living tissues and the osseointegration level in bone-implant surfaces. Furthermore, it would not interfere with the qualitative and comparative results because they were present for all models. Future fatigue life studies and clinical evaluations may complement the results or evaluating the different framework materials described for this technique.

## Conclusion

For this treatment modality, regardless the loading region, PEEK can be suggested as framework material to reduce the bone strain around the implants and the stress concentration in the bar structure. However, the use of PEEK increase the risk of prosthetic screws loosening and even fracture in comparison with metallic alloys.

## Author Contributions

**Conceptualization:** Regina Furbino Villefort, João Paulo Mendes Tribst, Amanda Maria de Oliveira Dal Piva, Alexandre Luiz Borges, Carlos Eduardo de Almeida Ferreira, Sandra Lúcia Ventorim von Zeidler.

**Data curation:** Regina Furbino Villefort, João Paulo Mendes Tribst, Alexandre Luiz Borges, Nívia Castro Binda, Carlos Eduardo de Almeida Ferreira, Sandra Lúcia Ventorim von Zeidler.

**Formal analysis:** Regina Furbino Villefort, João Paulo Mendes Tribst, Alexandre Luiz Borges, Nívia Castro Binda.

**Funding acquisition:** Regina Furbino Villefort, Amanda Maria de Oliveira Dal Piva, Sandra Lúcia Ventorim von Zeidler.

**Investigation:** Regina Furbino Villefort, João Paulo Mendes Tribst, Amanda Maria de Oliveira Dal Piva, Alexandre Luiz Borges, Carlos Eduardo de Almeida Ferreira, Marco Antonio Bottino, Sandra Lúcia Ventorim von Zeidler.

**Methodology:** Regina Furbino Villefort, João Paulo Mendes Tribst, Alexandre Luiz Borges, Nívia Castro Binda.

**Project administration:** Regina Furbino Villefort, Amanda Maria de Oliveira Dal Piva, Alexandre Luiz Borges, Nívia Castro Binda, Marco Antonio Bottino, Sandra Lúcia Ventorim von Zeidler.

**Resources:** Regina Furbino Villefort, João Paulo Mendes Tribst, Amanda Maria de Oliveira Dal Piva, Carlos Eduardo de Almeida Ferreira, Sandra Lúcia Ventorim von Zeidler.

**Software:** Regina Furbino Villefort, João Paulo Mendes Tribst, Amanda Maria de Oliveira Dal Piva, Alexandre Luiz Borges, Marco Antonio Bottino, Sandra Lúcia Ventorim von Zeidler.

**Supervision:** Regina Furbino Villefort, João Paulo Mendes Tribst, Amanda Maria de Oliveira Dal Piva, Alexandre Luiz Borges, Carlos Eduardo de Almeida Ferreira, Marco Antonio Bottino, Sandra Lúcia Ventorim von Zeidler.

**Validation:** Regina Furbino Villefort, João Paulo Mendes Tribst, Amanda Maria de Oliveira Dal Piva, Alexandre Luiz Borges, Carlos Eduardo de Almeida Ferreira, Marco Antonio Bottino, Sandra Lúcia Ventorim von Zeidler.

**Visualization:** Regina Furbino Villefort, João Paulo Mendes Tribst, Amanda Maria de Oliveira Dal Piva, Alexandre Luiz Borges, Nívia Castro Binda, Marco Antonio Bottino, Sandra Lúcia Ventorim von Zeidler.

**Writing – original draft:** Regina Furbino Villefort, João Paulo Mendes Tribst, Amanda Maria de Oliveira Dal Piva, Nívia Castro Binda, Sandra Lúcia Ventorim von Zeidler.

**Writing – review & editing:** Regina Furbino Villefort, João Paulo Mendes Tribst, Amanda Maria de Oliveira Dal Piva, Alexandre Luiz Borges, Nívia Castro Binda, Carlos Eduardo de Almeida Ferreira, Marco Antonio Bottino, Sandra Lúcia Ventorim von Zeidler.

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
