## [Decision Letter · Decision Letter 0]

30 Sep 2020

PONE-D-20-26196

Stress distribution on different bar materials in implant-retained palatal obturator

PLOS ONE

Dear Dr. Tribst,

Thank you for submitting your manuscript to PLOS ONE. After careful consideration, we feel that it has merit but does not fully meet PLOS ONE’s publication criteria as it currently stands. Therefore, we invite you to submit a revised version of the manuscript that addresses the points raised during the review process.

We look forward to receiving your revised manuscript.

Kind regards,

Antonio Riveiro Rodríguez, PhD

Academic Editor

PLOS ONE

"This research was supported by FAPES/CAPES

Grant: FAPES/CAPES Nº 10/2018 – PROFIX 2018

Process number: 83574662

Funding granted to: Regina Furbino Villefort ".

i) Please provide an amended statement that declares *all* the funding or sources of support (whether external or internal to your organization) received during this study, as detailed online in our guide for authors at http://journals.plos.org/plosone/s/submit-now.  Please also include the statement “There was no additional external funding received for this study.” in your updated Funding Statement.

ii) Please include your amended Funding Statement within your cover letter. We will change the online submission form on your behalf. 

3. Please upload a copy of Supporting Information Figures S1 - S12 which you refer to in your text on pages 17 - 18.

Reviewers' comments:

Reviewer's Responses to Questions

**Comments to the Author**

1. Is the manuscript technically sound, and do the data support the conclusions?

Reviewer #1: No

Reviewer #2: Partly

2. Has the statistical analysis been performed appropriately and rigorously? 

Reviewer #1: N/A

Reviewer #2: N/A

3. Have the authors made all data underlying the findings in their manuscript fully available?

Reviewer #1: Yes

Reviewer #2: Yes

4. Is the manuscript presented in an intelligible fashion and written in standard English?

Reviewer #1: Yes

Reviewer #2: Yes

5. Review Comments to the Author

Reviewer #1: This paper presents the results of the numerical investigations on the mechanical response of implant-retained obturator prostheses with bar-clip attachment and milled bars. The research topic is quite important and well suited in the journal scope. The results obtained are well interpreted. Moreover, structure of this manuscript and graphic quality of figures are acceptable.

There are some more comments from this reviewer that can help the Authors to improve the quality of the manuscript. Minor revisions are suggested. The detailed comments in the order of appearance in the text are summarized as follows.

The novelty of this manuscript has not been clearly specified. I found many similar numerical analyses in the literature. The elastic properties of materials are assumed basen on the literature. External hexagon implants (10 × 4.1 mm), previously modeled [25] were selected. An anteroposterior structure was used crossing the hemimaxillectomy defects similar to [11]. report.The value of simulated loads is based also on the literature. etc., etc., So, what is the novelty of this manuscript???

2. Definition of the materials is too general. In the Abstract the authors say that "PEEK, titanium and Co-Cr alloys" were considered. The grades of titanium and Co-Cr alloys and/or chemical composition must be clearly specified.

3. Line 30: "PEEK". Please define all abbreviations the first time they appear in the abstract, and the main text.

4. The method of determination of optimal mesh size is very mysterious. Line 122: "A convergence test of 10% determined [...]". 10% of what?

5. Conclusions are to general. The conclusions must be drawn appropriately based on the data presented.

Reviewer #2: Review comment

This manuscript entitled “Stress distribution on different bar materials in implant-retained palatal obturator” primarily aimed to evaluate the mechanical response of implant-retained obturator prostheses with bar-clip attachment and milled bars in three different materials: PEEK, titanium and cobalt-chromium alloys in different load incidences. The authors bring an interesting study. However, all the issues that listed below must be revised thoroughly before this manuscript being accepted for publication. I give a major revision for this manuscript.

Specific comments

Abstract

1. Abbreviations that exist in manuscript should be explained when they first appear, such as “PEEK” (Line 30). Please add the official explanation.

2. The detailed modeling method can be presented in the method session, the authors should give more information regarding the results, conclusion and application (if possible) of this study in this part.

Introduction

1. “PEEK has shown some advantages such as…, which are usual materials in restorative dentistry” (Line 65-68), some references should be added.

2. The novelty of this study should be further highlighted in the introduction part.

Methods

1. It is suggested that a flow chart for the pre-processing is needed in order to help readers better understand the methods.

2. Did this study apply any statistical methods to compare the differences between materials? If did, please specify it, and if not, please explain how the significant results came out without any statistical analysis.

Results

1. The results session should be rewritten based on the statistical analysis.

Discussion

1. “FEA” (Line 271), although I know the meaning of this abbreviation, the official explanations should be added.

2. “Computer-assisted implant planning has become …the most suitable mechanical results” (Line 282-285), this paragraph should be moved to the introduction part, as it seems to be the novelty of this study.

Conclusion

1. Are there any implications based on the results of this study? Please specify.

While written English is reasonably good, but please do check the language and grammar mistakes throughout the whole article to improve clarity.

6. PLOS authors have the option to publish the peer review history of their article (what does this mean?). If published, this will include your full peer review and any attached files.

Reviewer #1: No

Reviewer #2: **Yes: **Yaodong Gu

---

## [Author Response · Author response to Decision Letter 0]

1 Oct 2020

Reviewer #1: This paper presents the results of the numerical investigations on the mechanical response of implant-retained obturator prostheses with bar-clip attachment and milled bars. The research topic is quite important and well suited in the journal scope. The results obtained are well interpreted. Moreover, structure of this manuscript and graphic quality of figures are acceptable.

There are some more comments from this reviewer that can help the Authors to improve the quality of the manuscript. Minor revisions are suggested. The detailed comments in the order of appearance in the text are summarized as follows.

The novelty of this manuscript has not been clearly specified. I found many similar numerical analyses in the literature. The elastic properties of materials are assumed basen on the literature. External hexagon implants (10 × 4.1 mm), previously modeled [25] were selected. An anteroposterior structure was used crossing the hemimaxillectomy defects similar to [11]. report. The value of simulated loads is based also on the literature. etc., etc., So, what is the novelty of this manuscript???

R: First and foremost, we would like to thank the referee for the constructive criticisms, which have contributed to improve the content and the style of the manuscript. The reported parameters are used to ensure a well-controlled modelling process and it is common in studies like that. However, the finite element analysis of a hemimaxillectomy defect with this prosthesis design and different materials were never reported before in literature. Also, it is important to note that the implant model was previously reported but not in this kind of oral rehabilitation as you can see in the reference 25. In the same way, the reference 11 that was used as hemimaxillectomy defect reference is a case series, which have not evaluated the mechanical response. 

2. Definition of the materials is too general. In the Abstract the authors say that "PEEK, titanium and Co-Cr alloys" were considered. The grades of titanium and Co-Cr alloys and/or chemical composition must be clearly specified.

R: This information has been inserted in the text.

3. Line 30: "PEEK". Please define all abbreviations the first time they appear in the abstract, and the main text.

R: The PEEK definition has been inserted in the text.

4. The method of determination of optimal mesh size is very mysterious. Line 122: "A convergence test of 10% determined [...]". 10% of what?

R: 10% of difference in results between each calculation. This information has been inserted in the methods section.

5. Conclusions are to general. The conclusions must be drawn appropriately based on the data presented.

R: The conclusion section has been improved.

Reviewer #2: Review comment

This manuscript entitled “Stress distribution on different bar materials in implant-retained palatal obturator” primarily aimed to evaluate the mechanical response of implant-retained obturator prostheses with bar-clip attachment and milled bars in three different materials: PEEK, titanium and cobalt-chromium alloys in different load incidences. The authors bring an interesting study. However, all the issues that listed below must be revised thoroughly before this manuscript being accepted for publication. I give a major revision for this manuscript.

Specific comments

Abstract

1. Abbreviations that exist in manuscript should be explained when they first appear, such as “PEEK” (Line 30). Please add the official explanation.

R: The abbreviation has been explained in the text.

2. The detailed modeling method can be presented in the method session, the authors should give more information regarding the results, conclusion and application (if possible) of this study in this part.’

R: Thank you. The abstract section has been improved which more emphasis in the results and shorten modeling process.

Introduction

1. “PEEK has shown some advantages such as…, which are usual materials in restorative dentistry” (Line 65-68), some references should be added.

R: References were inserted in this statement.

2. The novelty of this study should be further highlighted in the introduction part.

R: An improvement in the study’s novelty has been performed in the line 73.

Methods

1. It is suggested that a flow chart for the pre-processing is needed in order to help readers better understand the methods.

R: A new figure 1 was inserted with a flow chart for the pre-processing step.

2. Did this study apply any statistical methods to compare the differences between materials? If did, please specify it, and if not, please explain how the significant results came out without any statistical analysis.

R: As the finite element method already provide a calculated results, there is no need to apply a posterior statistic test. In order to predict difference between the groups, the stress maps can be compared qualitatively and the stress peaks (Table 2) assumed as “significant different”, instead statically signiant difference, if this difference was higher than 10% (The mesh convergence, line 129. This result evaluation and interpretation is well defined and accepted in literature, as you observe in studies with similar methodology.

Results

1. The results session should be rewritten based on the statistical analysis.

R: Thank you for your suggestion, however it is not applicable for the present study.

Discussion

1. “FEA” (Line 271), although I know the meaning of this abbreviation, the official explanations should be added.

R: Thank you for your observation. The abbreviation has been explained in the text.

2. “Computer-assisted implant planning has become …the most suitable mechanical results” (Line 282-285), this paragraph should be moved to the introduction part, as it seems to be the novelty of this study.

R:This paragraph has been moved to the introduction section.

Conclusion

1. Are there any implications based on the results of this study? Please specify.

R: The conclusion has been modified to: “For this treatment modality, regardless the loading region, PEEK can be suggested as framework material to reduce the bone strain around the implants and the stress concentration in the bar structure. However, the use of PEEK increases the risk of prosthetic screws loosening and even fracture in comparison with metallic alloys”.

While written English is reasonably good, but please do check the language and grammar mistakes throughout the whole article to improve clarity.

R: The language has been reviewed.

---

## [Decision Letter · Decision Letter 1]

12 Oct 2020

PONE-D-20-26196R1

Stress distribution on different bar materials in implant-retained palatal obturator

PLOS ONE

Dear Dr. Tribst,

Thank you for submitting your manuscript to PLOS ONE. After careful consideration, we feel that it has merit but does not fully meet PLOS ONE’s publication criteria as it currently stands. Therefore, we invite you to submit a revised version of the manuscript that addresses the points raised during the review process.

Please, address the changes suggested by reviewer 2.

We look forward to receiving your revised manuscript.

Kind regards,

Antonio Riveiro Rodríguez, PhD

Academic Editor

PLOS ONE

Reviewers' comments:

Reviewer's Responses to Questions

**Comments to the Author**

1. If the authors have adequately addressed your comments raised in a previous round of review and you feel that this manuscript is now acceptable for publication, you may indicate that here to bypass the “Comments to the Author” section, enter your conflict of interest statement in the “Confidential to Editor” section, and submit your "Accept" recommendation.

Reviewer #1: All comments have been addressed

Reviewer #2: (No Response)

2. Is the manuscript technically sound, and do the data support the conclusions?

Reviewer #1: Yes

Reviewer #2: Yes

3. Has the statistical analysis been performed appropriately and rigorously? 

Reviewer #1: N/A

Reviewer #2: Yes

4. Have the authors made all data underlying the findings in their manuscript fully available?

Reviewer #1: Yes

Reviewer #2: Yes

5. Is the manuscript presented in an intelligible fashion and written in standard English?

Reviewer #1: Yes

Reviewer #2: Yes

6. Review Comments to the Author

Reviewer #1: (No Response)

Reviewer #2: Review comment

The authors have addressed the comments carefully. However, there are still some questions that must be revised before this manuscript being accepted for publication. I give a minor revision for this manuscript.

Specific comments

1. Please merge the fourth and fifth paragraphs of the Introduction session further.

2. The purpose of this study was to evaluate the mechanical response of implant-retained obturator prostheses with bar-clip attachment and milled bars in three different materials under two load incidences, however, from the conclusion (Abstract and Conclusions part), it seems that the authors only indicated some advantages and disadvantages of PEEK material, what about the other two materials? Any differences? Please specify.

3. Abbreviations that exist in main content of this manuscript only need to be explained when they first appear, Line 251 polyetheretherketone (PEEK), Line 292 finite element analysis (FEA), please modify.

7. PLOS authors have the option to publish the peer review history of their article (what does this mean?). If published, this will include your full peer review and any attached files.

Reviewer #1: No

Reviewer #2: **Yes: **Yaodong Gu

---

## [Author Response · Author response to Decision Letter 1]

14 Oct 2020

Reviewer #2: The authors have addressed the comments carefully. However, there are still some questions that must be revised before this manuscript being accepted for publication. I give a minor revision for this manuscript.

Specific comments

1. Please merge the fourth and fifth paragraphs of the Introduction session further.

R: These paragraphs were merged as suggested.

2. The purpose of this study was to evaluate the mechanical response of implant-retained obturator prostheses with bar-clip attachment and milled bars in three different materials under two load incidences, however, from the conclusion (Abstract and Conclusions part), it seems that the authors only indicated some advantages and disadvantages of PEEK material, what about the other two materials? Any differences? Please specify.

R: Yes. As we can observe in the results section. Since PEEK is the novelty, our conclusion was based on its mechanical response. However, the conclusion section has been improved.

3. Abbreviations that exist in main content of this manuscript only need to be explained when they first appear, Line 251 polyetheretherketone (PEEK), Line 292 finite element analysis (FEA), please modify.

R: The abbreviation explaining has been already presented in the first apparition of PEEK, in line 65. The first apparition of FEA was in line 79 and the abbreviation explaining has been already presented too.

---

## [Decision Letter · Decision Letter 2]

19 Oct 2020

Stress distribution on different bar materials in implant-retained palatal obturator

PONE-D-20-26196R2

Dear Dr. Tribst,

We’re pleased to inform you that your manuscript has been judged scientifically suitable for publication and will be formally accepted for publication once it meets all outstanding technical requirements.

Kind regards,

Antonio Riveiro Rodríguez, PhD

Academic Editor

PLOS ONE

Reviewers' comments:

Reviewer's Responses to Questions

**Comments to the Author**

1. If the authors have adequately addressed your comments raised in a previous round of review and you feel that this manuscript is now acceptable for publication, you may indicate that here to bypass the “Comments to the Author” section, enter your conflict of interest statement in the “Confidential to Editor” section, and submit your "Accept" recommendation.

Reviewer #2: All comments have been addressed

2. Is the manuscript technically sound, and do the data support the conclusions?

Reviewer #2: (No Response)

3. Has the statistical analysis been performed appropriately and rigorously? 

Reviewer #2: (No Response)

4. Have the authors made all data underlying the findings in their manuscript fully available?

Reviewer #2: (No Response)

5. Is the manuscript presented in an intelligible fashion and written in standard English?

Reviewer #2: (No Response)

6. Review Comments to the Author

Reviewer #2: (No Response)

7. PLOS authors have the option to publish the peer review history of their article (what does this mean?). If published, this will include your full peer review and any attached files.

Reviewer #2: **Yes: **Yaodong Gu

---

## [Editor Report · Acceptance letter]

21 Oct 2020

PONE-D-20-26196R2 

Stress distribution on different bar materials in implant-retained palatal obturator 

Dear Dr. Tribst:

I'm pleased to inform you that your manuscript has been deemed suitable for publication in PLOS ONE. Congratulations! Your manuscript is now with our production department. 

Kind regards, 

on behalf of

Dr. Antonio Riveiro Rodríguez 

Academic Editor

PLOS ONE